# Study protocol: azithromycin therapy for chronic lung disease of prematurity (AZTEC) - a randomised, placebo-controlled trial of azithromycin for the prevention of chronic lung disease of prematurity in preterm infants

John Lowe,[1] David Gillespie,[1] Marie Hubbard,[2] Lei Zhang,[3] Nigel Kirby,[1] Timothy Pickles,[1] Emma Thomas-Jones,[1] Mark A Turner,[4] Nigel Klein,[5] Julian R Marchesi,[6] Kerenza Hood,[1] Janet Berrington,[7] Sailesh Kotecha [3]

For numbered affiliations see end of article.

**Correspondence to**
Professor Sailesh Kotecha;
KotechaS@cardiff.ac.uk

## ABSTRACT

**Introduction** Chronic lung disease of prematurity (CLD), also known as bronchopulmonary dysplasia (BPD), is a cause of significant respiratory morbidity in childhood and beyond. Coupled with lung immaturity, infections (especially by *Ureaplasma* spp) are implicated in the pathogenesis of CLD through promotion of pulmonary inflammation. Azithromycin, which is a highly effective against *Ureaplasma* spp also has potent anti-inflammatory properties. Thus, azithromycin therapy may improve respiratory outcomes by targeting infective and inflammatory pathways. Previous trials using macrolides have not been sufficiently powered to definitively assess CLD rates. To address this, the azithromycin therapy for chronic lung disease of prematurity (AZTEC) trial aims to determine if a 10-day early course of intravenous azithromycin improves rates of survival without CLD when compared with placebo with an appropriately powered study.

**Methods and analysis** 796 infants born at less than 30 weeks' gestational age who require at least 2 hours of continuous respiratory support within the first 72 hours following birth are being enrolled by neonatal units in the UK. They are being randomised to receive a double-blind, once daily dose of intravenous azithromycin (20 mg/kg for 3 days, followed by 10 mg/kg for a further 7 days), or placebo. CLD is being assessed at 36 weeks' PMA. Whether colonisation with *Ureaplasma* spp prior to randomisation modifies the treatment effect of azithromycin compared with placebo will also be investigated. Secondary outcomes include necrotising enterocolitis, intraventricular/cerebral haemorrhage, retinopathy of prematurity and nosocomial infections, development of antibiotic resistance and adverse reactions will be monitored.

**Ethics and dissemination** Ethics permission has been granted by Wales Research Ethics Committee 2 (Ref 18/WA/0199), and regulatory permission by the Medicines and Healthcare Products Regulatory Agency (Clinical Trials Authorisation reference 21323/0050/001–0001). The study

## Strengths and limitations of this study

► This is a randomised, double-blind, placebo-controlled trial assessing an important outcome, and using an intervention which could be readily implemented in clinical practice.
► If the assumptions underlying the sample size calculation are valid, this study will be appropriately powered for the primary outcome.
► The study will also help address the role of pulmonary *Ureaplasma* spp colonisation in the development of chronic lung disease of prematurity.
► Universal administration of azithromycin may alter antibiotic resistance patterns, which are also being investigated in this study.
► Further work will be required to investigate the mechanistic action of azithromycin, including on the lung and gut microbiota.

is registered on ISRCTN (ISRCTN11650227). The study is overseen by an independent Data Monitoring Committee and an independent Trial Steering Committee. We shall disseminate our findings via national and international peer-reviewed journals, and conferences. A summary of the findings will also be posted on the trial website.

## INTRODUCTION

Chronic lung disease of prematurity (CLD), also known as bronchopulmonary dysplasia (BPD), is a major cause of neonatal death in infants born prematurely. It is clear that survivors of CLD have adverse respiratory outcomes in childhood and beyond.[1–3]

Despite the advances in neonatal care, rates of CLD have not markedly changed. This is largely because of the survival of the most immature infants born at the limits viability

(22 to 23 weeks' gestation), which means that the proportion of preterm infants born at 28 weeks or less of gestation has increased over the last 30 years.[4] Consequently, the etiological definition of CLD was modified to represent the structural immaturity of the lungs which are in the canalicular/saccular stages of development.[5 6] The pathophysiology is complex and results from continual lung injury caused by inflammatory processes coupled with repair/remodelling and resultant fibrosis.[7] In contrast, the historical picture of CLD was largely characterised by parenchymal tissue damage caused by volutrauma, barotrauma and by high levels of oxygen required to sustain life.[8]

Infections, both antenatal (eg, chorioamnionitis) and nosocomial, contribute to a cytokine-mediated inflammatory cascade which peaks between 7 to 10 days of life.[9 10] *Ureaplasma* spp (class Mollicutes) are the smallest free living organisms and have long been implicated in the pathogenesis of CLD.[11] These commensal bacteria form part of the normal vaginal bacteria microbiota, and thus are readily transmitted to the uterus via the cervix or acquired during vaginal delivery. Lacking a cell wall, they are readily transferred to the lung where they colonise the respiratory mucosa and directly or, in conjunction with aforementioned iatrogenic factors, indirectly promote inflammation largely through recruitment of neutrophils to the lung. Our recent systematic review and meta-analysis reported that *Ureaplasma* spp colonisation was strongly associated with diagnosis of CLD at 36 weeks' post-menstrual age (PMA; OR 2.22; 95% CI 1.42 to 3.47) which was largely independent of gestational age when investigated with meta-regression.[12]

Current treatments for the prevention of CLD are largely supportive and include optimising use of early non-invasive respiratory support and preventing infections. Use of systemic corticosteroids is considered in cases where the infant is unable to be weaned from invasive ventilation. However, although early low-dose regimens may be beneficial in a high-risk subpopulation, it is well-established that higher doses of corticosteroids are associated with adverse neurodevelopmental outcomes, including cerebral palsy.[13]

Due to the potential risks associated with use of corticosteroids, other treatments, or ideally preventive strategies, with improved safety profiles are urgently required for CLD. An interest in macrolide antibiotics, and the potential for improving lung outcomes via eradication of *Ureaplasma*, was established in the 1990s. However, early trials of erythromycin were underpowered and did not reduce the incidence or severity of CLD.[14] A study trialling clarithromycin treatment for 10 days reported a reduction in CLD (2.9% vs 36%) when compared with placebo but importantly only initiated delayed randomised therapy if the infant was culture-positive for *Ureaplasma*.[15] More contemporary data exists with azithromycin, which is an important adjunct therapy for numerous respiratory conditions, including cystic fibrosis and chronic obstructive pulmonary disease (COPD). Consistent with other macrolides, azithromycin inhibits bacterial protein synthesis through blinding to the 50S bacterial ribosomal subunit.[16] Moreover, azithromycin is attractive as it also uniquely exhibits well-characterised immunomodulatory effects through suppression of nuclear factor-kappaB, decreasing neutrophilic pulmonary inflammation by limiting the production of pro-inflammatory cytokines such as interleukin (IL)-6 and IL-8.[17] Furthermore, since azithromycin is concentrated in leukocytes, it may be actively transported and released at the site of infection.[18]

Several proof of concept and dose-finding studies have already been completed. The meta-analysis by Nair and colleagues showed that azithromycin treatment is associated with a 14% reduction in the combined outcome of CLD/death when compared with placebo.[19] Dose-finding work in respect of *Ureaplasma* spp eradication has been undertaken in a series of studies by Viscardi and colleagues.[20–22] Importantly, all previous studies did not note any serious adverse reactions to azithromycin and add to the existing safety information.[23] Antimicrobial resistance among *Ureaplasma* remains low.[24]

Given the Nair meta-analyses and lack of adequately powered studies, and given the reported association of presence of pulmonary inflammation and *Ureaplasma* with development of CLD, there is a compelling case for a definitive, adequately powered study to investigate the effectiveness of azithromycin on reducing rates of CLD in preterm infants. The azithromycin therapy for chronic lung disease of prematurity (AZTEC) trial is determining if a 10-day course of intravenous azithromycin improves rates of survival without CLD at 36 weeks' PMA when compared with placebo.

## METHODS AND ANALYSIS
### Primary objective
The primary objective of the AZTEC trial is to assess the effectiveness of a 10-day course of azithromycin on improving survival without physiologically-defined CLD in infants born at <30 weeks' gestational age.

### Design
AZTEC is a double-blind, randomised, placebo-controlled trial. The total samples size is 796 preterm-born infants; each are randomised individually (including infants from a multiple birth). Follow-up is undertaken at 36 weeks' PMA to assess CLD status at 36 weeks' PMA, with final follow-up a discharge from hospital.

### Setting
Infants are being enrolled from UK tertiary neonatal units, which are designated Level III (regional neonatal intensive care units), and followed up at their local hospital if transferred. Infants are identified by the study team on admission and screened against the inclusion/exclusion criteria.

### Inclusion criteria
1. Gestational age ≤29 weeks+6 days (including infants born as one of a multiple birth)

2. Infants who receive respiratory support for at least 2 continuous hours' duration during the first 72 hours of life (intubated, or by non-invasive mechanical ventilation, including continuous positive airway pressure and high-flow nasal cannula, or a combination thereof).
3. Presence of an indwelling intravenous line for drug administration.
4. Written informed parental/guardian consent within 72 hours of birth.
5. Anticipating administration of first dose within 72 hours of birth at the latest (optimally targeting within 24 hours after birth for inborn and 48 hours for outborn infants).
6. Reasonable expectancy to complete 10 days of trial treatment while resident at the recruiting site.
7. Inborn, or born at site within the recruiting site's neonatal network where follow-up will be possible.
8. In the opinion of the local principal investigator (PI), reasonable prospect of survival past the first 72 hours of life.

### Exclusion criteria
1. Exposure to another systemic macrolide antibiotic (not maternal)
2. Presence of major surgical or congenital abnormalities (not including patent ductus arteriosus or patent foramen ovale)
3. Contraindication of azithromycin as specified in the summary of product characteristics (SPC)
4. Participation in other interventional trial that precludes participation in AZTEC

### Trial intervention
The investigational medicinal product (IMP) is manufactured and QP released by Saint Mary's Pharmaceutical Unit (SMPU), Cardiff, UK (MIA(IMP)35929).

The dosing schedule is 20 mg/kg (10 mL/kg) azithromycin for 3 days, followed by 10 mg/kg (5 mL/kg) for 7 days or placebo (10 days total). All doses are given via intravenous infusion (central or peripheral line) over a period of at least 1 hour. Azithromycin is most likely to have an effect when administered early to establish a sufficient concentration to eliminate *Ureaplasma*; a 20 mg/kg dose for 3 days has recently been shown to be highly effective.[22] Treatment for a further 7 days is justified to treat the rise in pulmonary inflammation which peaks between 7 and 10 days after birth.[9 25] Sites have therefore been asked to target initiation of trial treatment at the earliest opportunity (and within 72 hours after birth at the latest).

### Blinding
IMP is supplied as a patient pack of 12 blinded vials. The vial blinding method uses a custom cardboard carton sourced by SMPU, as used previously in a similar trial design.[26] Labelling was performed by SMPU as per a randomisation list provided by the Centre for Trials Research (CTR), Cardiff University.

**Table 1** Severity-based criteria for diagnosis of CLD at 36 weeks post-menstrual age

| Received respiratory support and/or supplementary oxygen for **more than** 28 days, cumulatively, and the following: | |
| --- | --- |
| Mild CLD | ► Breathing room air |
| Moderate CLD | ► Require <30% oxygen (or low flow 0.01 to 1.0 L/min), **not** receiving any respiratory support |
| Severe CLD | ► Require ≥30% oxygen (or low flow ≥1.1 L/min), **still** receiving respiratory support (ventilation, CPAP, high-flow oxygen) |

CLD, chronic lung disease of prematurity; CPAP, continuous positive airway pressure.

### Active arm
The active arm comprises of commercially-available 500 mg vials of azithromycin powder for solution for infusion (Aspire Pharma Ltd). As per the summary of product characteristics (https://www.medicines.org.uk/emc/product/1276/smpc), 4.8 mL sterile water is used to reconstitute the powder to obtain a clear, colourless solution of 100 mg/mL. The administration concentration of 2 mg/mL is obtained by withdrawing 1 mL of solution from the vial and adding to 50 mL of an acceptable diluent.

### Control arm
The placebo arm initially comprised of an empty, sterile vial manufactured to the same specification as the active product (released under GMP standards) and provided to SMPU by Aspire Pharma Ltd for blinding and packaging. The same preparation steps are followed as per the active arm to produce a solution matching in appearance to the active arm.

## OUTCOMES
### Primary outcome
The primary outcome is a combined outcome of CLD (moderate-severe) and mortality at 36 weeks' PMA (or discharge, if sooner). The definition of CLD severity is based on consensus criteria,[27] table 1.

Infants meeting the initial diagnosis of moderate CLD will undergo a physiological test to confirm their oxygen requirement (figure 1). This physiological definition, initially developed by Quine and colleagues,[28] has been widely used in clinical trials of neonatal lung disease.[29]

Secondary outcomes will include
1. Mortality rate by 36 weeks PMA.
2. CLD severity by 36 weeks of age or discharge.
3. Number of days of respiratory support/oxygen dependency.
4. Development of complications of prematurity.
   a. Nosocomial infection.

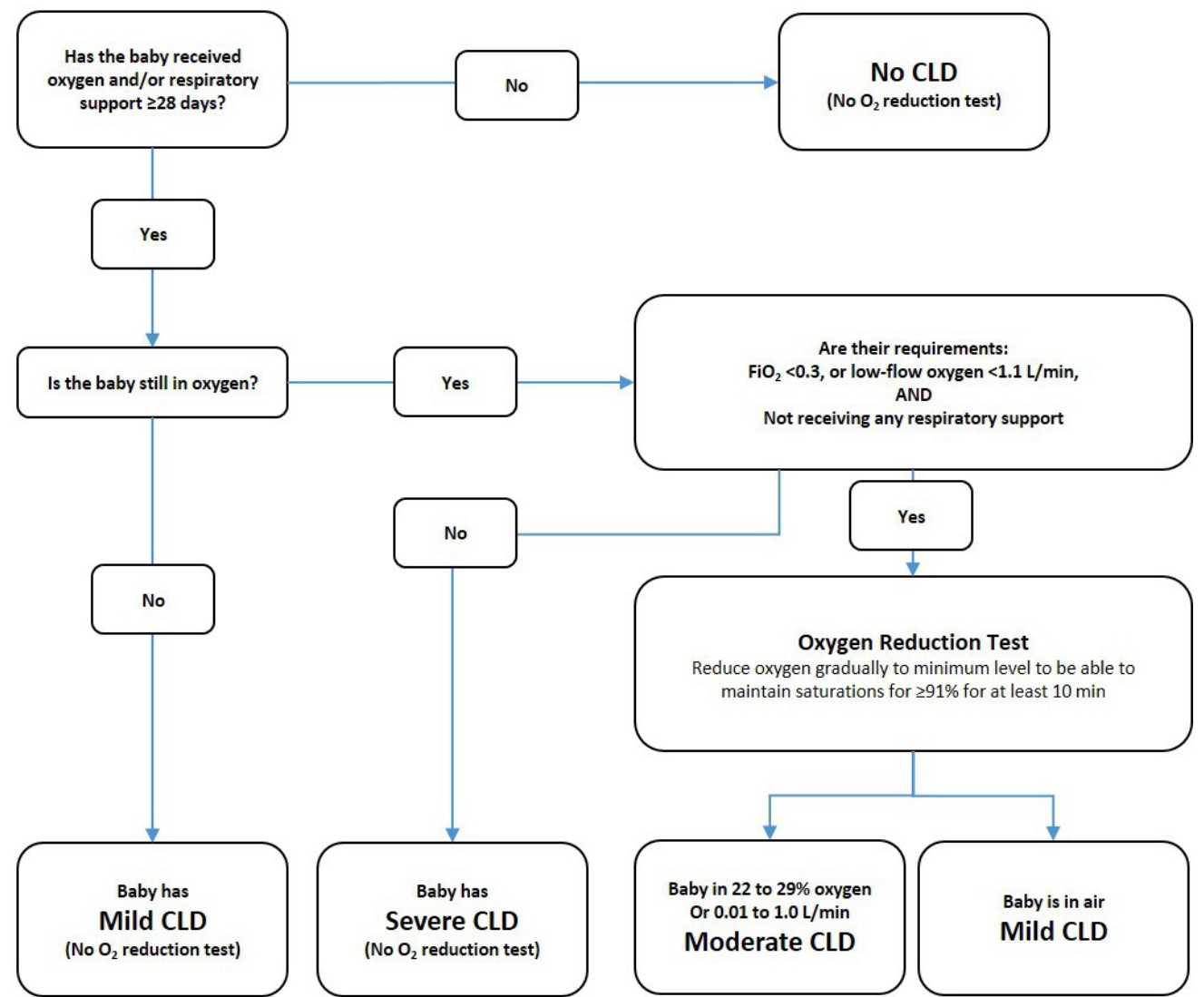

**Figure 1** Flow diagram for assessment of CLD severity in the AZTEC trial. Modified from the original https://www.npeu.ox.ac. uk/downloads/files/baby-oscar/protocol/Baby-OSCAR_Protocol_v6_171116.pdf. AZTEC, azithromycin therapy for chronic lung disease of prematurity; CLD, chronic lung disease of prematurity; FIO$_2$, fraction of inspired oxygen.

  b. Severe intraventricular haemorrhage (grade III/ IV).
  c. Necrotising enterocolitis (Bell stage II and above).
  d. Treatment for retinopathy of prematurity.
  e. Treatment for patent ductus arteriosus.
  f. Liver and renal function.
5. Serious adverse events/reactions.
6. Resistance to macrolides among microbes isolated from stool samples.

## Trial procedures
### Site selection and training
Site selection is based on receipt of an expression of interest via the National Institute for Health Research (NIHR) networks and personal contact directly with UK Level III neonatal units. A registration questionnaire captures key information about the site, trial team and any concerns around delivery of the protocol in the context of routine practice. Where feasible, a preliminary visit is made by the Trial Manager to present the study background and protocol. The Trial Management Group approved the centres selected to participate.

Formal initiation visits are held at each site to train the local trial team in trial-specific procedures. This is supplemented with comprehensive guidance documents for discrete elements of the study. Specific training in preparation and administration of the IMP is cascaded down by delegated members of the local study team to cot-side nurses; this includes a video produced by JL with support from neonatal nurses and the medical illustration department at Cardiff and Vale University Health Board (https://youtu.be/Xto6n5qFuYQ). The local clinical trials pharmacy team are also provided with trial-specific training in IMP storage, accountability and reconciliation procedures during the initiation process. All local study teams have undertaken Good Clinical Practice training commensurate with their roles and responsibilities.

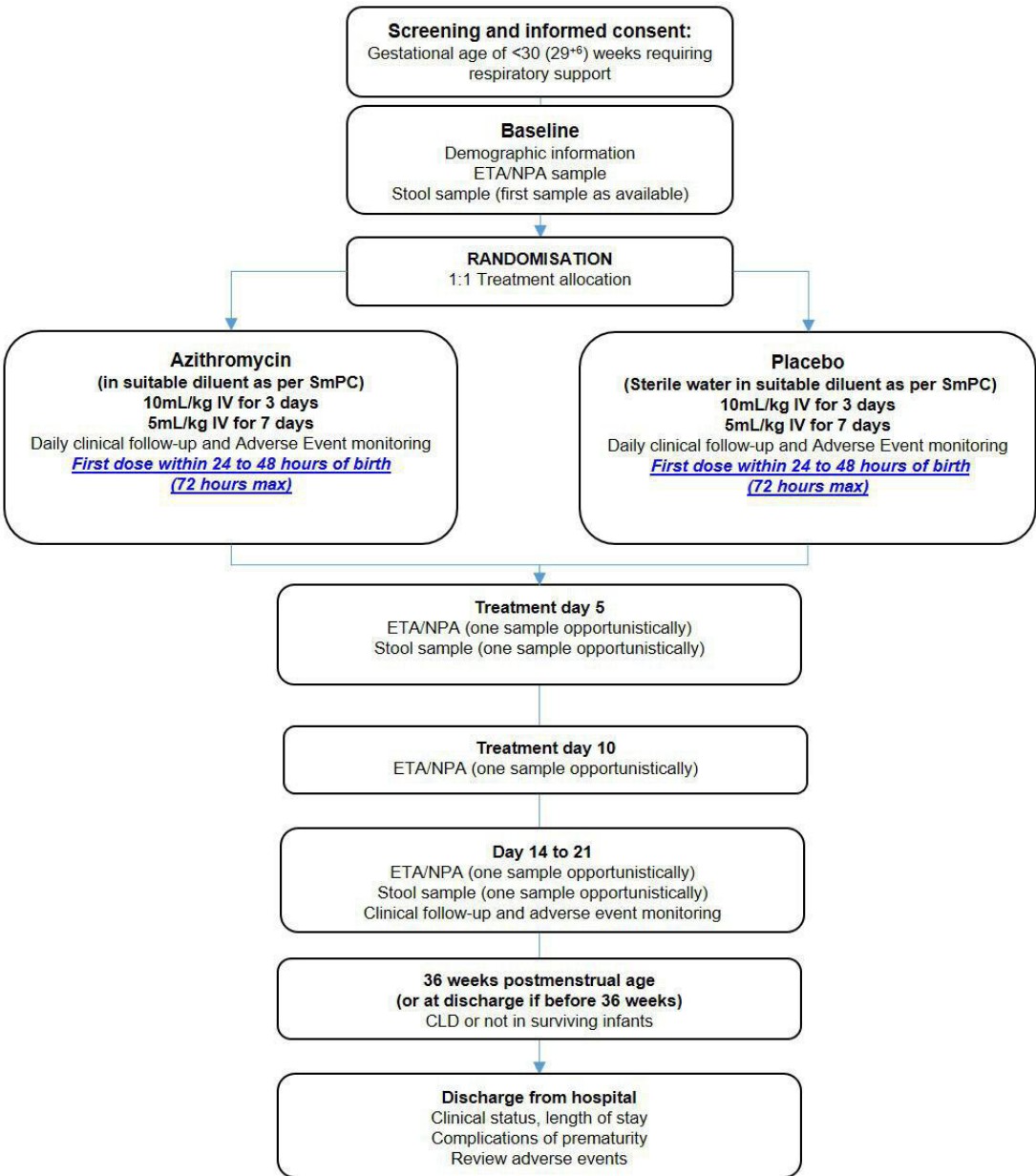

**Figure 2** Flow diagram for AZTEC study procedures and follow-up. AZTEC, azithromycin therapy for chronic lung disease of prematurity; ETA, endotracheal aspirates; NPA, nasopharyngeal aspirates.

## Participant recruitment

The trial design is summarised in figure 2. Recruitment is anticipated to take 30 months, with 25 sites participating (an average of 1.5 infants per site, per month over the course of recruitment). Progress of the trial can be followed at www.aztec-trial.uk

## Screening and consent

It is anticipated that all infants who meet the gestational age criteria for the trial (ie, born <30 weeks' gestational age) are registered on the anonymised screening log. Reasons for exclusion, and reasons for not randomising otherwise eligible infants (eg, declined consent) are being recorded.

A member of the local site team identifies potential eligible preterm babies antenatally with mothers with threatened preterm labour or babies admitted to the neonatal unit and the parent(s) receive a verbal description of the trial. Should the parent(s) express an interest, the ethically-approved information sheet and consent form are provided (online supplemental file). Eligibility is confirmed by a medically qualified member of the trial team. Parents are given sufficient time to read the information and to ask questions during a further consultation with the study team. If the parents are willing to participate, they are asked to sign the consent form which is countersigned by the PI or a delegated member of the study team who engaged the parents in the informed consent discussion. Consent for use of samples in ancillary studies, and for future follow-up contact are optional. The right to withdraw from the study at any time without

any affecting their baby's clinical care will be clearly communicated.

## Randomisation

Randomisation to the intervention is performed following confirmation of eligibility and completion of the informed consent process. The randomisation list prepared by an independent CTR statistician uses a 1:1 allocation ratio with fixed block length. Allocation to azithromycin or placebo is blinded such that the allocation will not be known to clinicians, the baby's family or the trial outcome assessors. The list is uploaded to a validated, user-tested, web-based system (Sortition, Oxford University Innovation Ltd). Randomisation is performed by a member of the local study team using a unique username and password. The system performs treatment allocation by issuing a random four-digit IMP pack ID (matching the IMP supplied to each site), and also a participant ID in a standardised format. A member of the site study team retrieves the appropriate pack of IMP from the storage location and ensures this is appropriately prescribed on the infant's prescription chart. Unblinding, if required, may also be performed by the site PI (or designee) using the web-based system.

## Data collection and sampling
### Clinical assessments

Following documentation of eligibility, the focus of the baseline data collection is on maternal history and antenatal information, and details around the infant's birth.

IMP administration, sample collection, use of intravenous antibiotics and details of any positive blood/cerebrospinal fluid cultures are recorded on a daily basis until the infant is 21 days post randomisation. Liver and kidney function will be monitored through collection of standard laboratory values. Should the infant be transferred from the AZTEC recruiting site to another hospital for continuation of care, the receiving centre continues with

follow-up assessments to support collection of primary and secondary outcomes.

Infants remain in follow-up for safety and outcome purposes until 36 weeks' PMA, or are discharged home sooner (the last time point for recording new adverse reactions).

### Data collection

All data are recorded on the trial electronic Case Report Form. Accumulating data are regularly monitored and queries raised with sites should values be missing or otherwise erroneous (eg, validation against pre-specified ranges for laboratory values).

### Sampling

Sites are collecting serial respiratory secretions and stool samples in a pragmatic, opportunistic manner around nominal time points of baseline, day 5, day 10 and days 14 to 21 post-randomisation. These procedures are timed to mirror standard care to minimise disturbing trial infants, where possible. Endotracheal aspirates are obtained if the infant is intubated, and nasopharyngeal aspirates are collected if they are not. Stool samples are collected opportunistically whenever the infants open their bowels. All samples are refrigerated until shipment at the earliest opportunity overnight to the central laboratory where they are processed according to a standardised procedure and stored at −80°C pending analysis. A schematic of the sampling plan is show in figure 3.

## Analysis
### Sample size

Relevant interventional studies to prevent development of CLD as an outcome (including studies using macrolides) in preterm infants were reviewed. In general, national and international studies consistently show rates of survival without CLD of 50% to 60% (those with lower rates are due to highly selected groups of sicker participants) for

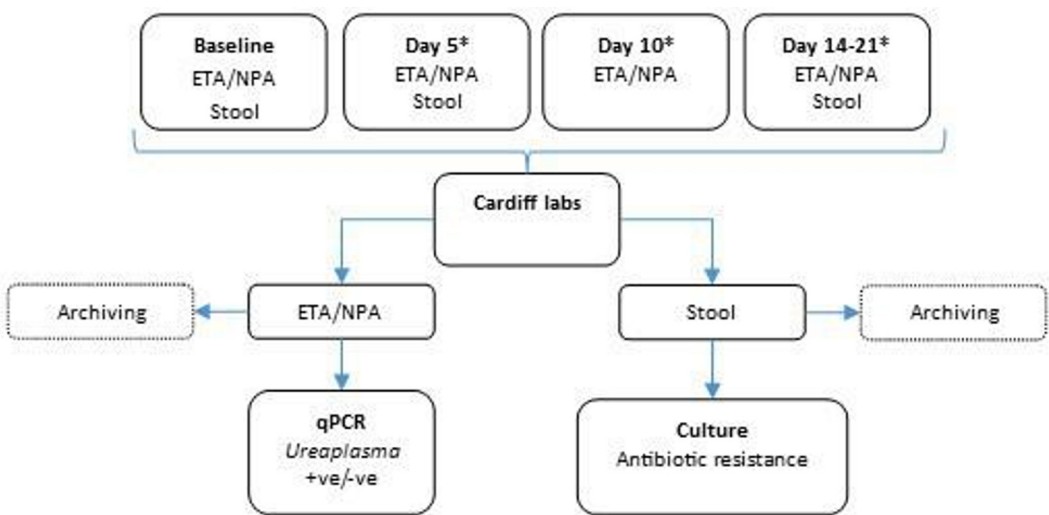

**Figure 3** AZTEC biological sampling plan* day 4 to 6 is acceptable for the day 5 samples. Day 8 to 12 is acceptable for the day 10 samples. Day 14 to 21—one sample of each type during this period. AZTEC, azithromycin therapy for chronic lung disease of prematurity; CLD, chronic lung disease of prematurity; ETA, endotracheal aspirates; NPA, nasopharyngeal aspirates.

studies with similar inclusion criteria to AZTEC. Thus, adopting a conservative approach using 50% survival without developing CLD is reasonable. Similarly, the absolute differences in effect size ranges from 10% to 20% in most studies. The study is powered to an improvement of 12% (50% to 62%) in survival without CLD with a power of 0.90 and significance level of 5% requiring recruitment of 796 subjects which is highly feasible as shown by several similar UK studies in this population. Since the primary outcome will involve formal assessment with an oxygen challenge test in both tertiary units and in step-down units, a dropout rate of 10% has been estimated.

## Statistical analysis

We are conducting an internal pilot involving the first five sites to be activated, and for a period of 12 months (9 months of recruitment and 3 months of follow-up). A pilot report will be produced detailing information on recruitment rate, consent rates, treatment compliance and primary outcome completeness. The report will be presented to the funder and the AZTEC independent committees to help inform any adaptations necessary to facilitate moving forward with the main trial. No formal interim analysis is planned.

The trial will be analysed and reported using the 'Consolidated Standards of Reporting Trials' (CONSORT)[30] and the International Conference on Harmonization (ICH) E9 guidelines.[31] A separate and full statistical analysis plan has been developed prior to database lock. The analysis plan will be reviewed by the AZTEC independent committees.

The principle of intention-to-treat will be applied as far as practically possible, including all participants in the primary analysis in the group to which they were randomly allocated. All analyses will use a 5% level of statistical significance and 95% CI will be presented throughout. The primary outcome will be analysed using multilevel logistic regression, within a multiple imputation framework. The analysis will adjust for covariates balanced at randomisation and account for clustering of both multiple births and participants within centres. The imputation model will use the treatment arm and gestational age variables, as well as whether or not the participant was transferred from their recruiting site prior to the primary outcome assessment. Furthermore, a series of sensitivity analyses, within the multiple imputation framework, will be used to assess the robustness of conclusions.

The presence of multiple births within the same pregnancy will be explored by including this as an additional level in the model, and this will be used as the primary analytical approach should the model converge. Dichotomous secondary outcomes will be analysed using the same approach. Number of days of respiratory support will be analysed as a time to event outcome allowing for competing risk of death. These models will also attempt to account for any clustering effects of multiple births within the same mother. Statistical tests will not be used

on safety and tolerability outcomes. These outcomes will use descriptive statistics only.

To explore the extent to which there may be a differential treatment effect by presence of *Ureaplasma* spp, the model fitted for the primary analysis will be extended by including a main and treatment group interaction term for *Ureaplasma* spp colonisation at baseline. Since patterns of *Ureaplasma* colonisation may vary,[22 32] the approach to this analysis may be modified in respect of emerging trends identified the placebo group samples. The final analytical approach will be agreed prior to database lock. Estimates from the statistical models (main effects and interaction terms) will be presented alongside 95% CI and p values.

There are no formal stopping rules but the safety data will be regularly reviewed by the Trial Management Group, and at least annually by the independent Data Monitoring Committee.

## Microbiology

Respiratory samples are being processed via centrifugation to separate the supernatant and obtain a cell pellet from which DNA will be extracted. Identification of *Ureaplasma urealyticum*, *Ureaplasma parvum* and *Mycoplasma hominis* is being performed using multiplex quantitative PCR (qPCR) assay as previously described.[33 34] All DNA extractions and qPCR assays are being performed in collaboration with Central Biotechnology Services (Cardiff University, UK) according to standard operating procedures and GCP to minimise contamination.

Stool samples are being cultured aerobically and non-aerobically, and in the presence and absence of azithromycin to assess baseline and new development of azithromycin-resistant organisms.

## DISCUSSION

As the survival of preterm infants increases, decreasing morbidity associated with prematurity, including CLD, is becoming increasingly important. AZTEC is an example of evaluating drug development[35] in infants as a large, pragmatic and adequately powered trial of a macrolide antibiotic to prevent development of CLD with widespread support by the neonatal community.[36 37] The importance of improving outcomes for preterm infants at risk of developing CLD is highlighted by the potential impacts on later pulmonary function, especially the prospect of early decline in lung function and early-onset COPD.[38] Moreover, the evidence base to treat graduates of the neonatal unit when they present with symptoms in childhood is poor;[39] largely because the mechanisms of prematurity-associated respiratory morbidity remain poorly understood.[40]

Whether modification of pulmonary inflammation and treatment of *Ureaplasma* infection would reduce rates of CLD has been debated for three decades. Use of postnatal corticosteroids is associated with long-term adverse effects; however, the safety profile of azithromycin is superior

and is unlikely to affect neurological outcomes. Plans to follow-up neurodevelopment and respiratory outcomes at 1 and 2 years of age are in place. This study includes ascertainment of whether each infant is colonised with *Ureaplasma* spp and randomisation of treatment to azithromycin or placebo. Given the time lag between randomisation and the results of tests for *Ureaplasma* spp, the randomisation will not be balanced for *Ureaplasma* status. Nevertheless, this study design will allow an assessment of the causative role of *Ureaplasma* in the development of CLD.

If effectiveness of azithromycin therapy is demonstrated, integration into standard care would be relatively straightforward since the majority of candidates for treatment would have intravenous access for 7 to 10 days. Establishing whether azithromycin alters antibiotic resistance patterns will provide important information to the clinician who may have concerns around adding an adjunct treatment to a wide array of antimicrobials already given to this vulnerable group of patients. Although AZTEC will not specifically address mechanistic actions of azithromycin, including its effect on the lung and gut microbiota, samples will be banked to permit such work in the near future using established[25] and novel methodologies such as metagenomics.[41]

In summary, the AZTEC trial is addressing an important therapeutic need in an area where treatment based on high-quality evidence is severely lacking. Should antimicrobial resistance not be affected, azithromycin therapy should be appealing to clinicians due to the anti-inflammatory properties and proven ability to eradicate *Ureaplasma*. Treatment may yield fewer intensive care days and fewer discharges on home oxygen, resulting in considerable reduction in costs to the National Health Service (NHS), as well as decreasing the significant burden on parents. On the other hand, should the treatment not be clinically effective, the neonatal research community can turn their attention to alternative options for combating CLD. Identification of an effective treatment to prevent CLD would be of exceptional value to the NHS, infants and their families.

## ETHICS AND DISSEMINATION

The current version of the AZTEC protocol is 3.0, dated 20 June 2020. Ethics permission has been granted by the Wales Research Ethics Committee 2 (Ref 18/WA/0199), and regulatory permission by the Medicines and Healthcare Products Regulatory Agency (CTA reference 21323/0050/001–0001). NHS permission has been granted by the Health Research Authority (HRA) and capacity and capability confirmed by each individual NHS organisation. The study is registered on EudraCT (2018-001109-99), ISRCTN (ISRCTN11650227) and on the NIHR portfolio (CPMS 39385). Cardiff University is the Sponsor (resgov@cardiff.ac.uk), and were not involved in the preparation of this manuscript or the decision to submit. Personal data is held with the explicit consent

of participants, independently of study data. The CTR has policies and procedures relating to data requests: https://www.cardiff.ac.uk/centre-for-trials-research/about-us/data-requests. We shall disseminate our findings via national and international peer-reviewed journals, and conferences. A summary of the findings will also be posted on the trial website.

Oversight of the study is being performed by an independent Data Monitoring Committee (comprising two expert neonatologists, and an expert statistician) and an independent Trial Steering Committee (comprising two expert neonatologists, an expert statistician and a lay representative). Appointments to these committees were made with approval of the NIHR HRA and meeting a being held at least annually.

## Patient and public involvement

Parental input was obtained during the grant application on the design and conduct of the study. Parent representatives also reviewed public-facing information (eg, information sheets and consent form) and are members of the Trial Management Group. An independent parent representative is a member of the Trial Steering Committee.

**Author affiliations**
[1]Centre for Trials Research, College of Biomedical and Life Sciences, Cardiff University, Cardiff, UK
[2]Neonatal Intensive Care Unit, University Hospitals of Leicester NHS Trust, Leicester, UK
[3]Department of Child Health, School of Medicine, Cardiff University, Cardiff, United Kingdom
[4]Institute of Translational Medicine, University of Liverpool, Liverpool, United Kingdom
[5]GOS Institute of Child Health, University College London, London, UK
[6]School of Biosciences, Cardiff University, Cardiff, UK
[7]Neonatal Intensive Care Unit, Newcastle Upon Tyne Hospitals NHS Foundation Trust, Newcastle Upon Tyne, UK

**Acknowledgements** We are very grateful for the support provided from both the independent Data Monitoring Committee and Trial Steering Committee. The Centre for Trials Research receives core funding from the Welsh Government via Health & Care Research Wales and Cancer Research UK. We would like to acknowledge and thank the expert help we received from many parents and nursing staff to develop the initial application and protocol. In particular, we would like to thank the babies and their parents who have started to be enrolled to AZTEC.

**Contributors** SK is the Chief Investigator who developed the research question and secured funding together with JL, MT and JB, with subsequent input from JRM, NK, KH, DG and ETJ. JL and SK wrote the protocol with input from all authors. JL is the Trial Manager, managed by ETJ, Senior Trial Manager, and is responsible for coordinating the operational delivery of the protocol and recruitment. DG and TP are the Trial Statisticians and authored the statistical analysis plan. JL and NKi led the development of the case report forms; NKi supervises data management. LZ is the Laboratory Analyst; expert microbiology advice is provided by NK and JRM. SK, MT, MH and JB provide expert clinical input; MH provides nursing and clinical expertise. JL wrote the first draft of the manuscript supervised by SK. All authors provided critical review and final approval of the manuscript.

**Funding** This work is supported by the National Institute of Health Research Health Technology Assessment programme (Ref 16/111/106). The study duration is 01 January 2019 to 30 September 2022.

**Competing interests** None declared.

**Patient and public involvement** Patients and/or the public were involved in the design, or conduct, or reporting, or dissemination plans of this research. Refer to the Methods section for further details.

**Patient consent for publication**  Not required.

**Provenance and peer review**  Not commissioned; externally peer-reviewed.

**ORCID iD**

Sailesh Kotecha http://orcid.org/0000-0003-3535-7627

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
