## [Reviewer comments · BMJ Open]

ARTICLE DETAILS

TITLE (PROVISIONAL)	Study Protocol: Azithromycin Therapy for Chronic Lung Disease of Prematurity (AZTEC): A randomised, placebo-controlled trial of azithromycin for the prevention of chronic lung disease of prematurity in preterm infants
AUTHORS	Lowe, John; Gillespie, David; Hubbard, Marie; Zhang, Lei; Kirby, Nigel; Pickles, Timothy; Thomas-Jones, Emma; Turner, Mark; Klein, Nigel; Marchesi, Julian; Hood, KerENZA; Berrington, Janet; Kotecha, Sailesh

VERSION 1 – REVIEW

REVIEWER	Rose Viscardi University of Maryland School of Medicine Baltimore, MD 21201 USA
REVIEW RETURNED	28-Jun-2020

GENERAL COMMENTS	This is a well-written manuscript describing a study protocol for a double-blind RCT of a 10 day course of IV azithromycin to prevent moderate to severe BPD in preterm infants <30 weeks gestation. Overall, the details of the trial are well-described and rationale for the 10 day course of therapy and early initiation of therapy are provided. However, there are a number of details listed in the SPIRIT Checklist that could be better addressed in the manuscript. These include: 1. Stopping rules for an individual subject or for the trial. Although previous trials of azithromycin in the preterm population did not identify adverse events associated with the drug, safety concerns may become apparent in this larger trial. For instance, will the study drug be discontinued if a subject develops an arrhythmia?2. Description of processes for data quality such as out of range values, missing data, etc.3. In statistical analysis plan, how they will handle missing data such as failure to conduct room air challenge for primary outcome (e.g. multiple imputation)4. More details on composition of the DSMB, frequency of meetings, including where details of the DSMB charter can be found.5. Although the investigators state there will be an interim analysis after the 12 month pilot study, they do not address whether any other interim analysis is planned or stopping rules. Another consideration is the classification of subjects as Ureaplasma positive or negative based on a single baseline ET or NP aspirate. In the experience of the Phase II RCT (reference #22), our group conducted, for some subjects, the 2 ET/NPA samples obtained at baseline were negative, but follow-up
--

	samples were positive. Subjects who were positive at any timepoint were considered positive for analyses. The AZTEC investigators might consider that approach. Furthermore, we observed that subjects in our trial who were TA Ureaplasma positive had worse outcomes than those infants who were TA negative or never intubated. Our group developed the PK/PD of azithromycin for the 20 mg/kg x 3 days dose (reference #21), but azithromycin drug levels with an additional 10 mg/kg x7 days is not known. The AZTEC investigators might consider obtaining residual blood specimens for drug levels in a small subset of subjects for a PK analysis.
--	--

REVIEWER	Gaetano Chirico Just retired from NICU Spedali Civili Brescia, Italy
REVIEW RETURNED	04-Jul-2020

GENERAL COMMENTS	This is a very well designed study protocol aimed to evaluate the use of azithromycin for the prevention of chronic lung disease of prematurity. Methods have been adequately reported, including Outcome, Sample size, Recruitment, Allocation implementation, Blinding, Data collection methods and retention, Statistical methods, Population analysed. Only Access to data should be better described. Minor comments: A typo should be corrected (pag 3 line 43 cannalicular: canalicular) The evaluation of Liver function test should be underlined. It is correctly stated that the use of postnatal corticosteroids is associated with long term adverse effects, however, the quoted review suggest that low total doses of dexamethasone may nonetheless be beneficial for a subpopulation of high-risk infants.
---

VERSION 1 – AUTHOR RESPONSE

Reviewer: 1

This is a well-written manuscript describing a study protocol for a double-blind RCT of a 10 day course of IV azithromycin to prevent moderate to severe BPD in preterm infants <30 weeks gestation. Overall, the details of the trial are well-described and rationale for the 10 day course of therapy and early initiation of therapy are provided.

- We thank the reviewer for this positive feedback

However, there are a number of details listed in the SPIRIT Checklist that could be better addressed in the manuscript. These include:

1. Stopping rules for an individual subject or for the trial. Although previous trials of azithromycin in the preterm population did not identify adverse events associated with the drug, safety concerns may become apparent in this larger trial. For instance, will the study drug be discontinued if a subject develops an arrhythmia?

- We thank this reviewer for this important point. Through the involvement of our trial management group, adverse reactions are being reviewed on a monthly basis by experienced neonatologists. Moreover, these data form an integral part of the reporting to our Independent Data Monitoring Committee who perform a review at least annually. Although discontinuation of the IMP in the event of a serious adverse reaction is at the discretion of the local investigator, this is the primary course of

action and each site was advised as such during study-specific pharmacovigilance training. We have noted this at the end of the “statistical analysis” section, page 12 of the revised tracked change manuscript.

2. Description of processes for data quality such as out of range values, missing data, etc.

- We have added the following to the Data Collection and Sampling section on page 10 of the revised manuscript “Data Collection: All data are recorded on the trial electronic Case Report Forms. Accumulating data are being regularly monitored and queries raised with sites should values be missing, or otherwise erroneous (e.g. validation against pre-specified ranges for laboratory values).”

3. In statistical analysis plan, how they will handle missing data such as failure to conduct room air challenge for primary outcome (e.g. multiple imputation)

- The reviewer raises an important point since some infants may not undergo the test if transferred to another site where permissions are not in place or if there is not clinical capacity to accommodate this test when it is due. As noted in the manuscript, we have inflated our sample size to accommodate 10% missing primary outcome data (including missing room air challenge). We have updated the statistical analysis section to state “The primary outcome will be analysed using multilevel logistic regression, within a multiple imputation framework”. Moreover, we shall conduct a series of sensitivity analyses to explore the robustness of the conclusions.

4. More details on composition of the DSMB, frequency of meetings, including where details of the DSMB charter can be found.

- We have updated the “Ethics and dissemination” section on page 12 to outline the composition of both independent committees, and the frequency of meetings.

5. Although the investigators state there will be an interim analysis after the 12 month pilot study, they do not address whether any other interim analysis is planned or stopping rules.

- We have updated the statistical analysis section 11 of the revised tracked manuscript to state that although an internal pilot is being conducted to assess the feasibility of the trial, there are no formal interim analyses planned, and there are no formal stopping rules

6. Another consideration is the classification of subjects as Ureaplasma positive or negative based on a single baseline ET or NP aspirate. In the experience of the Phase II RCT (reference #22), our group conducted, for some subjects, the 2 ET/NPA samples obtained at baseline were negative, but follow-up samples were positive. Subjects who were positive at any timepoint were considered positive for analyses. The AZTEC investigators might consider that approach. Furthermore, we observed that subjects in our trial who were TA Ureaplasma positive had worse outcomes than those infants who were TA negative or never intubated.

- This comment, which we are obviously aware of, is very helpful and we shall ensure that the patterns of expression of Ureaplasma will be included in our analyses and we may well approach our expert colleague, who has positively reviewed this manuscript, in the future. We have added Ref #22 and stated that different patterns of pulmonary Ureaplasma colonisation have been observed which will be included in the analyses when the main SAP is fully developed. Furthermore, any data that emerge from our blinded analysis of the placebo group samples will inform our statistical analyses plan of how to best include the patterns of colonisation in our final analyses.

7. Our group developed the PK/PD of azithromycin for the 20 mg/kg x 3 days dose (reference #21), but azithromycin drug levels with an additional 10 mg/kg x7 days is not known. The AZTEC investigators might consider obtaining residual blood specimens for drug levels in a small subset of subjects for a PK analysis.

- We agree that additional PK studies would provide useful data. However, the UK approach tends to be pragmatic clinical trials thus AZTEC is funded to provide evidence for effectiveness of the treatment in a pragmatic trial, without any opportunities to include mechanistic work within our study protocol.

Reviewer: 2

Please leave your comments for the authors below This is a very well designed study protocol aimed to evaluate the use of azithromycin for the prevention of chronic lung disease of prematurity. Methods have been adequately reported, including Outcome, Sample size, Recruitment, Allocation implementation, Blinding, Data collection methods and retention, Statistical methods, Population analysed. Only Access to data should be better described.

- We thank the reviewer for their positive feedback. We shall include the data access policy in the final paper and ensure that availability is according to the prevailing governance requirements at the time of publication.

Minor comments:

A typo should be corrected (page 3 line 43 cannalicular: canalicular) The evaluation of Liver function test should be underlined.

- We have corrected the typo as indicated, thank you. We have outlined that “Data collection and sampling- Clinical assessments” that “liver and kidney function will be monitored through collection of standard laboratory values”

It is correctly stated that the use of postnatal corticosteroids is associated with long term adverse effects, however, the quoted review suggest that low total doses of dexamethasone may nonetheless be beneficial for a subpopulation of high-risk infants.

- We have balanced this statement to include “although early low-dose regimens may be beneficial in a high-risk subpopulation”.

VERSION 2 – REVIEW

REVIEWER	Rose Marie Viscardi, M.D. University of Maryland School of Medicine Baltimore, MD, USA
REVIEW RETURNED	03-Aug-2020

GENERAL COMMENTS	The authors have adequately addressed in the revision the concerns raised in the Review of the first submission. I look forward to the results of this important trial.
--

REVIEWER	Chirico, Gaetano Department of Neonatology and NICU, Spedali Civili, Brescia, Italy
REVIEW RETURNED	09-Aug-2020

GENERAL COMMENTS	In my opinion, the Authors have given satisfactory responses to the comments
--